# Mitochondrial Dynamics in Placenta-Derived Mesenchymal Stem Cells Regulate the Invasion Activity of Trophoblast

**DOI:** 10.3390/ijms21228599

**Published:** 2020-11-14

**Authors:** Jin Seok, Sujin Jun, Jung Ok Lee, Gi Jin Kim

**Affiliations:** 1Department of Biomedical Science, CHA University, Seongnam 13488, Korea; jjin8977@gmail.com (J.S.); junsj00@gmail.com (S.J.); 2Department of Anatomy, Korea University College of Medicine, Seoul 02841, Korea; misocell@korea.ac.kr

**Keywords:** placenta-derived mesenchymal stem cells, trophoblast, invasion, mitochondrial dynamics, mitophagy

## Abstract

Mitochondrial dynamics are involved in many cellular events, including the proliferation, differentiation, and invasion/migration of normal as well as cancerous cells. Human placenta-derived mesenchymal stem cells (PD-MSCs) were known to regulate the invasion activity of trophoblasts. However, the effects of PD-MSCs on mitochondrial function in trophoblasts are still insufficiently understood. Therefore, the objectives of this study are to analyze the factors related to mitochondrial function and investigate the correlation between trophoblast invasion and mitophagy via PD-MSC cocultivation. We assess invasion ability and mitochondrial function in invasive trophoblasts according to PD-MSC cocultivation by quantitative reverse transcription polymerase chain reaction (qRT-PCR) and extracellular flux (XF) assay. Under PD-MSCs co-cultivation, invasion activity of a trophoblast is increased via activation of the Rho signaling pathway as well as Matrix metalloproteinases (MMPs). Additionally, the expression of mitochondrial function (e.g., reactive oxygen species (ROS), calcium, and adenosine triphosphate (ATP) synthesis) in trophoblasts are increased via PD-MSCs co-cultivation. Finally, PD-MSCs regulate mitochondrial autophagy factors in invasive trophoblasts via regulating the balance between PTEN-induced putative kinase 1 (PINK1) and parkin RBR E3 ubiquitin protein ligase (PARKIN) expression. Taken together, these results demonstrate that PD-MSCs enhance the invasion ability of trophoblasts via altering mitochondrial dynamics. These results support the fundamental mechanism of trophoblast invasion via mitochondrial function and provide a new stem cell therapy for infertility.

## 1. Introduction

Successful embryo implantation requires the dynamic invasion ability of trophoblasts. Trophoblast cells provide nutrients to the embryo and develop into a large part of the placenta. Trophoblasts are composed of several cell types, including cytotrophoblasts, syncytiotrophoblasts, and intermediate trophoblasts. Particularly, when cytotrophoblasts, which are located in the inner layer of trophoblasts, make contact, the trophoblast begins to rapidly proliferate and secrete proteolytic enzymes to break the extracellular matrix (ECM) for invasion [1,2]. There have been many reports on the mechanisms of invasion and trophoblast regulatory factors, including matrix metalloproteinases (MMPs) and the Rho family (e.g., rho-associated protein kinase 1 (ROCK1), phospho-focal adhesion kinase (p-FAK) and Rho A/B/C) [3,4]. However, the abnormal invasion of trophoblasts results in embryo implantation failure and abnormal placental development, resulting in obstetric diseases such as intrauterine growth retardation (IUGR) and preeclampsia [5]. Therefore, several studies are underway to improve the invasion ability of trophoblast cells.

Several recent studies reported that mesenchymal stem cells (MSCs) improved the migration and invasion ability of trophoblast cells by upregulating their proliferation. Huang et al., found that umbilical cord-derived mesenchymal stem cells (UCMSCs) improved both the migration and invasion abilities of trophoblasts by upregulating Human chorionic gonadotropin (hCG), phosphatidylinositol-glycan biosynthesis class F (PIGF), and Endoglin [6]. Regarding our previous reports, MSCs and extracts derived from the normal term placenta enhanced the invasion ability of trophoblasts via altering human leukocyte antigen G (HLA-G) expression [7]. Many researchers have demonstrated that MSC-derived exosomal miRNAs regulate the invasion and migration of trophoblasts via anti-inflammatory factors [8,9]. However, research to improve trophoblast invasion remains insufficient.

Mitochondria are essential for generating energy and controlling oxidative stress due to cellular processes, including invasion, but these organelles are sensitive to both oxidative stress and environmental toxicants [10,11]. Kim and their colleagues reported that moderate reactive oxygen species (ROS) levels support cell proliferation and migration. However, excessive ROS accumulation promotes sever cellular damage and triggers apoptosis [12]. Zsengeller et al., demonstrated that mitochondrial dysfunction induces an overwhelming increase in ROS levels and contributes to pathogenesis, including preeclampsia and miscarriage [13]. Additionally, several studies demonstrated that excessive ROS accumulation inhibits the invasion ability of trophoblasts via downregulated forkhead box protein O1 (FOXO1)P and integrin b3 [14,15].

Mitochondrial quality control systems are involved in maintaining cellular homeostasis through mitochondrial autophagy and mitochondrial biogenesis [15,16]. Mitochondrial autophagy, known as mitophagy, a major homeostatic mechanism, maintains the balance between mitochondrial fission and fusion [17,18]. Moreover, it involves the active and selective destruction of mitochondria by persistent hypoxic conditions [19,20]. Additionally, mitochondrial dysfunction, including imbalance between mitochondrial fission and fusion, can induce diseases related to trophoblast dysfunction [21,22]. PTEN-induced putative kinase 1 (PINK1) is widely known to protect cells from stress-induced mitochondrial damage and activate the parkin RBR E3 ubiquitin protein ligase (PARKIN) protein, which binds depolarized mitochondria [23]. The cytosolic E3 ubiquitin ligase PARKIN plays a role in recognizing proteins on the mitochondrial outer membrane and mediates the removal of damaged mitochondria via autophagy [24,25,26]. The PINK1–PARKIN pathway is associated with the selective autophagy of damaged mitochondria. Recently, Chen et al., reported that PINK1 phosphorylates Mitofusin-2 (MFN2) and PARKIN, bring about mitochondrial dysfunction [27]. However, the study on the effects of mitochondrial dysfunction including mitochondrial autophagy-related mechanisms during the invasion of trophoblasts remains unclear.

Therefore, the purpose of this study is to analyze alterations in mitochondrial dynamics in invasive trophoblasts and research the importance of cocultivation with placenta-derived mesenchymal stem cells (PD-MSCs) for the relationship between trophoblast invasion and mitophagy.

## 2. Results

### 2.1. PD-MSCs Enhance the Invasion Ability of Trophoblasts via MMP-2/9 Activity and the Rho Family Pathway

To investigate the cellular metabolism of invasive trophoblasts, we collected invaded cells by using a Trans-well insert system. Shown in Figure 1A, there were remarkably more invasive trophoblast cells among the placenta derived mesenchymal stem cell (PD-MSC) cocultivation group than among trophoblasts cultured alone. To quantify the invaded cells, we stained the invaded cells with hematoxylin. The proportion of stained cells was significantly increased in the PD-MSC cocultivation group compared with trophoblasts cultured alone (Figure 1B, * *p* < 0.05). Also, we confirmed that effect of PD-MSCs cocultivation in invasion ability of other trophoblastic cell lines. Shown Appendix A, PD-MSCs cocultivation induced the invasion ability of other trophoblastic cells including BeWo and JEG3 (Appendix A). The proportion of stained cells was remarkably increased in the PD-MSCs cocultivation group compared with other trophoblastic cells cultured alone (Appendix A). Based on previous reports, we selected factors related to invasion processing and analyzed their levels in invasive trophoblast cells by quantitative reverse transcription polymerase chain reaction (qRT-PCR). Hypoxia inducible factor 1 alpha (HIF1α) is a transcription factor known to regulate cell migration as well as cellular metabolism [28,29,30]. HIF1a mRNA expression at 24 h and 48 h was significantly increased in invasive trophoblasts cocultured with PD-MSCs compared with trophoblasts cultured alone (Figure 1C, * *p* < 0.05). Matrix metalloproteinase-2/-9 (MMP-2/-9) are involved in breakdown of the extracellular matrix (ECM) during trophoblast invasion. The mRNA expression of MMP-2/9 at 12 h, 24 h and 48 h was significantly increased in invasive trophoblasts cocultured with PD-MSCs compared with trophoblasts cultured alone (Figure 1D,E, * *p* < 0.05).The MMP-2 expression in supernatant were significantly increased in trophoblasts cocultured with PD-MSCs compared with trophoblasts cultured alone (Appendix A, * *p* < 0.05).

Additionally, we selected adhesion molecules involved in invasion and migration e.g., Focal adhesion kinase (*FAK*), Rho A, Rho-associated coiled-coil-containing protein kinase 1 (*ROCK1*), and Ras-related C3 botulinum toxin substrate 1 (*Rac1*) and assessed their expression. The mRNA expression of *FAK* and *Rho A* was significantly increased in invasive trophoblasts cocultured with PD-MSCs at 12 h and 24 h (Figure 1F,G, * *p* < 0.05). Interestingly, the mRNA expression of *Rock* and *Rac1* at 12 h, 24 h and 48 h was remarkably increased in invasive trophoblasts cocultured with PD-MSCs compared with trophoblasts cultured alone (Figure 1H,I, * *p* < 0.05). These findings indicated that PD-MSC cocultivation increased the invasion ability of trophoblasts via activation of MMP-2/9 expression and Rho family signaling.

### 2.2. PD-MSCs Induce Changes in the Mitochondrial Function of Invasive Trophoblasts

To analyze the mitochondrial function of trophoblasts after invasion, we investigated the levels of oxidative stress and antioxidant factors in invasive trophoblasts with and without PD-MSC cocultivation by qRT-PCR. Heme oxygenase-1/-2 (*HO-1/-2*) and Superoxide dismutase 1 (*SOD1*) are the early factors to respond immediately to oxidative stress, and rapidly beak down factor. Thus, the mRNA expression of *HO-1/-2*, which is related to oxidative stress, at 12 h, 24 h and 48 h was slightly increased in invasive trophoblasts cocultured with PD-MSCs compared with trophoblasts cultured alone (Appendix A, * *p* < 0.05). The mRNA expression of *SOD1*, an antioxidant enzyme, was slightly decreased in invasive trophoblasts cocultured with PD-MSCs compared with trophoblasts cultured alone (Appendix A). Although rapid break-down factors (i.e., HO-1/-2 and SOD1), these data showed slight differences according to PD-MSCs cocultivation. These data indicated that PD-MSCs cocultivation induced moderate ROS levels via activated cellular migration.

To assess mitochondrial energy metabolism, Adenosine triphosphate (ATP) production in the whole-cell lysates of invasive trophoblast cells with and without PD-MSC cocultivation was assayed. Shown by the Enzyme-linked immunosorbent assay (ELISA) results, ATP production was significantly decreased in trophoblasts cocultured with PD-MSCs compared with trophoblasts cultured alone at 12 h, 24 h and 48 h (Figure 2A, * *p* < 0.05).Interestingly, ATP binding cassette subfamily B member 10 (*ABCB10*) mRNA expression was considerably increased in invasive trophoblasts cocultured with PD-MSCs compared with trophoblasts cultured alone at 12 h, 24 h and 48 h (Figure 2B, * *p* < 0.05). Moreover, the mRNA expression of both Inositol trisphosphate receptor (*IP3R*) and mitochondrial calcium uniporter (*MCU*), which are related to ATP production, was significantly increased in invasive trophoblast cells cocultured with PD-MSCs at 12 h, 24 h and 48 h (Figure 2C,D, * *p* < 0.05). These data indicated that PD-MSC cocultivation increased ATP production compared with that in trophoblasts cultured alone by regulating calcium channel expression and mild oxidative stress.

### 2.3. PD-MSCs Regulated Glycolysis in Invasive Trophoblasts

To analyze alterations in glycolysis after trophoblast invasion, trophoblast cells were seeded in the wells of a 24-well Trans-well system with membrane inserts, and PD-MSCs were seeded in a cell culture plate. After the insert with the invaded cells was harvested, glycolysis was measured in invasive trophoblast cells of live condition by treatment with three chemicals, glucose, oligomycin and 2-Deoxy-D-Glucose (2-DG), as shown in Figure 3A. The extracellular acidification rate (ECAR) during chemical treatment was measured by determining changes in pH. The phenotype after the analysis is shown in Figure 3B. Invasive trophoblast cells cocultured with PD-MSCs showed high metabolic activity. We analyzed glycolysis in invasive trophoblast cells by quantification. The basal ECAR of invasive trophoblast cells was analyzed early in the analysis. The basal ECAR was increased in invasive trophoblast cells cocultured with PD-MSCs (Figure 3C). After glucose treatment, the ECAR was increased in invasive trophoblast cells cocultured with PD-MSCs at 24 h and 48 h (Figure 3D). Interestingly, the ECAR of invasive trophoblast cells cocultured with PD-MSCs indicated decreased glycolytic capacity and increased glycolytic reserve (Figure 3E,F, * *p* < 0.05). These data indicate that trophoblast cells cocultured with PD-MSCs require energy-dependent glycolysis during invasion, unlike trophoblasts cultured alone.

### 2.4. PD-MSCs Changed the Mitochondrial Oxygen Consumption of Invasive Trophoblasts

The same cell culture conditions used to analyze glycolysis were used to analyze mitochondrial stress levels after trophoblast invasion. Mitochondrial stress levels were analyzed by treatment with three chemicals, oligomycin, phenylhydrazone (FCCP) and rotenone/AA, as shown in Figure 4A. After chemical treatment to inhibit mitochondrial function, the Oxygen consumption rate (OCR) was measured by determining changes in the pH levels of the culture media. The cell phenotype after the analysis is presented in Figure 4B. Occurring at the 24 h time point, the invasive trophoblast cells cocultured with PD-MSCs showed an energetic phenotype. Additionally, the invasive trophoblast cells cocultured with PD-MSCs exhibited a glycolytic phenotype at the 48-h time point. Additionally, we analyzed the basal OCR in invasive trophoblast cells with and without PD-MSC cocultivation. The OCR was lower in invasive trophoblast cells cocultured with PD-MSCs than in invasive trophoblast cells without PD-MSC cocultivation at 12 h and 24 h (Figure 4C). Additionally, ATP production was lower than that in invaded cells without PD-MSC cocultivation (Figure 4D, * *p* < 0.05). However, the maximal respiration and spare capacity of mitochondria were increased compared with those of the invasive trophoblast cells without PD-MSC cocultivation (Figure 4E,F, * *p* < 0.05). These data indicate that PD-MSC cocultivation had a mild effect on the mitochondrial oxygen consumption of invasive trophoblast cells but not trophoblast cells cultured alone.

### 2.5. PD-MSCs Trigger Mitochondrial Autophagy in Invasive Trophoblasts

We next examined whether PD-MSCs affect the mitochondrial autophagy pathway or trophoblasts during invasion. First, the mitochondrial DNA (mtDNA) copy number was analyzed to assess mitochondrial fission and fusion in trophoblast cells. The mtDNA copy number was significantly lower in invasive trophoblast cells cocultured with PD-MSCs than in trophoblast cells cultured alone (Figure 5A, * *p* < 0.05). Second, we analyzed the mRNA expression of genes related to mitochondrial fission and fusion by qRT-PCR. Both Dynamin 1 like protein (*Drp1*) and Optic atrophy 1 (*Opa1*), which are GTPases located at the mitochondrial membrane, regulate mitochondrial fission and fusion. *Drp1* and *Opa1* mRNA expression levels were significantly increased in invasive trophoblasts cocultured with PD-MSCs compared with trophoblasts cultured alone at 12 h, 24 h and 48 h (Figure 5B,C, * *p* < 0.05). Third, we analyzed the expression of genes related to mitochondrial autophagy (mitophagy) in invasive trophoblasts with and without PD-MSC cocultivation by qRT-PCR. The parkin RBR E3 ubiquitin protein ligase (*PARKIN*) is a well-known regulator that affects processes in mitophagy, such as cell survival, by suppressing both mitochondria-dependent and mitochondria-independent apoptosis. Additionally, the *PINK1* regulates mitophagy and protects cells from stress-induced mitochondrial dysfunction. As shown by the qRT-PCR results, the mRNA expression of *PINK1* and *PARKIN* was significantly increased in invasive trophoblasts cocultured with PD-MSCs compared with trophoblasts cultured alone (Figure 5D,E, * *p* < 0.05). These data suggest that PD-MSC cocultivation reduced mitochondrial damage in invasive trophoblast cells caused by excess energy consumption. Additionally, this data means that PD-MSC cocultivation protected against cell damage via the mitophagy pathway during trophoblast invasion.

## 3. Discussion

PD-MSCs have several advantages, including their high capacity for self-renewal, multipotent nature, and strong immunomodulatory effects, and secrete many more kinds of growth factors, cytokines, and chemokines than other MSCs [31,32]. Several researchers have demonstrated that human placental mesenchymal stem cells (hPMSCs) secrete various cytokines and growth factors. Chen and colleagues also demonstrated that hepatocyte growth factors from hPMSCs enhanced the invasion ability of trophoblasts via cyclic adenosine monophosphate (cAMP) production [33]. Moreover, our previous reports suggested that PD-MSCs can enhance the invasion activity of trophoblasts via upregulating MMP-2/9 activity and activating immunosuppression and human leukocyte antigen G (HLA-G) [7]. Here, we showed that PD-MSCs enhanced the invasion ability of trophoblasts via upregulation of hypoxia inducible factor 1 alpha(HIF1-a) and activation of matrix metalloproteinase (MMP-2/9) and the Rho signaling pathway. Additionally, we hypothesized that cocultivation with PD-MSCs would change the mitochondrial function of invasive trophoblasts.

Regarding mitochondria, ATP production by glycolysis and mitochondrial respiration plays an essential role in the metabolism of high-energy cells. Generally, metabolic cells constantly transform ingested nutrients into a supply of glucose, glutamine and lipids to balance the metabolic needs of both differentiated and proliferating cells [34,35]. Additionally, the movement of calcium (Ca^2+^) through the endoplasmic reticulum (ER)/mitochondria supplements the energy consumed by high-energy cells. The interaction between mitochondrial calcium uniporter (MCU) and inositol 1,4,5-trisphosphate (IP3R) is involved in ATP production via oxidative phosphorylation and plays a role in cytosolic calcium buffering [36,37,38]. Tarasov et al., reported that mitochondrial Ca^2+^ accumulation is mediated by MCU and, thus, required for normal glucose sensing by cells [39]. Recently, higher oxidative phosphorylation (OXPHOS) and ATP production/consumption were indicated in invasive cells compared with less invasive cells [40]. However, the levels of ATP and Gunanosine-5’-triphosphate (GTP) produced in the mitochondria of invasive cells are still unclear. Therefore, we analyzed the mitochondrial metabolic profile, including glycolysis and mitochondrial stress, using Seahorse XF24 metabolic flux analysis and assessed this issue through determining the mRNA expression of genes involved in mitochondrial ATP metabolism. Based on our data, PD-MSCs co-cultivation increased ATP consumption by invasive trophoblast cells compared to control trophoblast cells by changing metabolism (Figure 3 and Figure 4). Interestingly, the expression of ATP-binding cassette sub-family B member 10(ABCB10), which is required for ATP synthesis, was significantly increased in invasive trophoblasts cocultured with PD-MSCs. Additionally, we confirmed that PD-MSCs significantly increased the activation of Ca^2+^ transporters (i.e., MCU and IP3R) for ATP generation in invasive trophoblasts compared with control trophoblasts (Figure 2, * *p* < 0.05). These data demonstrate that PD-MSCs could maintain balance between ATP consumption and synthesis in invasive trophoblasts during the invasion process.

Various oxidative stresses occur in the cellular energy metabolism and the factors associated with ROS, and antioxidant enzymes were very rapidly dynamics in the metabolic process [41,42,43]. Hence, because oxidative stress and energy generation/consumption are important factors in trophoblast survival, we analyzed oxidative stress levels in invasive trophoblasts with and without PD-MSC cocultivation. Hurd, T.R. et al., reported that increased intracellular ROS levels by changes in the balance between Heme oxygenase -1(HO-1) and HO-2 promote cell migration and adhesion via NADPH oxidase (NOX) signaling [44]. Also, mtROS signaling increases MMP9 mRNA stability and affects the invasiveness of cells [45]. Considering our data, moderated ROS levels by PD-MSCs cocultivation induced higher MMP activities in invasiveness. Additionally, Liu and colleagues demonstrated that secreted paracrine factors protect against oxidative stress [14]. These data are similar to our data that PD-MSCs induced the mRNA expression of HO-1/-2 in invasive trophoblasts compared with control trophoblasts. Consequently, these data indicate that oxidative stress generated by PD-MSC cocultivation decreases stress caused by active metabolism to a level below that at which cell death is promoted.

Autophagy has three types, including micro-/macro-autophagy and chaperone-mediated autophagy (CMA). Several studies were reported that autophagy signaling may contribute to enhanced cellular invasion processing via an upregulated Nuclear factor kappa light chain enhancer of activated B cells (NF-κB) gene [46,47]. Hence, we focused on mitochondrial autophagy (mitophagy) of macro-autophagy regarding mitochondrial function. Mitophagy, a quality control system in cellular mitochondria, is an essential pathway that protects mitochondria from damage [48]. We analyzed the effect of PD-MSCs on mitochondrial quality control systems involved in cellular homeostasis (e.g., the PINK1-PARKIN signaling pathway) in invasive trophoblast cells. Recently, Heeman, B. et al., reported that PINK1 modulates the mitochondrial network, energy maintenance and calcium homeostasis [49]. Thus, our data indicated that PD-MSCs cocultivation triggered mitochondrial autophagy of trophoblast cells for activated invasion ability.

These reports might be consistent with our data showing the decrease in the mtDNA copy number in invasive trophoblasts cocultivated with PD-MSCs. The mRNA expression of dynamin-1-like protein (Drp1) and opa1 mitochondrial dynamin like GTPase (Opa1), which are involved in mitochondrial fission and fusion, was increased in invasive trophoblasts cocultured with PD-MSCs. Additionally, the mRNA expression of PINK1 and PARKIN was significantly increased in invasive trophoblasts cocultured with PD-MSCs. Our data indicate that PD-MSCs can regulate the mitochondrial dynamics of invasive trophoblasts via mitophagy during invasion (Figure 6). Shown in Figure 6, PD-MSCs trigger trophoblast invasion via upregulating HIF1α while activating MMPs and the Rho family signaling pathway (1). Activated signalizing by PD-MSCs regulate the mitochondrial function of invasive trophoblasts through increased mRNA expression of both MCU and IP3R, and ABCB10 (2). These events regulate mitochondrial fragmentation via maintaining the balance between fission and fusion in trophoblasts (3). Finally, mitochondrial autophagy in trophoblasts changed via upregulating PINK1 and PARKIN expression, resulting in increased invasion ability of trophoblasts (4). Taken together, PD-MSCs promote trophoblast invasion via regulating mitochondrial dynamics, including ATP production and mitochondrial autophagy. These results suggest the fundamental mechanisms of trophoblast invasion via mitochondrial function and provide a new stem cell therapy for implantation.

## 4. Materials and Methods

### 4.1. HTR-8/SVneocells and Human Placenta-Derived Mesenchymal Stem Cells Culture

The human placenta at the early stage of pregnancy-derived HTR-8/SVneo trophoblastic cells was provided by Dr. Charles H. Graham (Queen’s University, Canada). HTR-8/SVneo cells were cultured in Roswell Park Memorial Institute (RPMI)-1640 medium (HyClone, GE Healthcare Life Sciences, Seoul, Korea) supplemented with 5% fetal bovine serum (FBS; Gibco-BRL, Rockville, MD, USA) and 1% penicillin/streptomycin (pen/strep; Gibco-BRL, Rockville, MD, USA) at 37 °C in an incubator with a humidified atmosphere of 5% CO_2_.

Human PD-MSCs were isolated from normal term placentas of women after approval from the Institutional Review Board of CHA General Hospital, Seoul, Korea (IRB 07-18) as described previously [50]. PD-MSCs were cultivated in α-minimum essential medium (α-MEM; HyClone, GE Healthcare Lifesciences, Seoul, Korea) supplemented with 10% FBS (Gibco-BRL), 1% pen-strep (Gibco-BRL), 25 μg/mL human Fibroblast Growth Factor 4 (hFGF-4; PeproTech, Inc., Rocky Hill, NJ, USA) and 1 μg/mL heparin (Sigma-Aldrich, St. Louis, MO, USA) at 37 °C in an incubator with a humidified atmosphere of 5% CO_2_. A 6-well Trans well system (with an insert with a 8-μm pore size; Falcon, BD Biosciences, Franklin Lakes, NJ, USA) was used to analyze the invaded HTR-8/SVneo cells. HTR-8/SVneo cells (1 × 10^5^/well) were seeded in the upper chamber of the Trans well system with 2 mL of medium, and PD-MSCs (2.5 × 10^5^/well) were seeded in the lower chamber with 0.7 mL of serum-free medium, after which cells were incubated for 12 h, 24 h, and 48 h. Invading cells that had attached to the bottom of the filter were harvested at each time point.

### 4.2. Quantitative Real-Time Polymerase Chain Reaction

Total RNA was isolated from invaded HTR-8/SVneo cells by using a TRIzol reagent (Ambion, Boston, MA, USA) following the manufacturer’s instructions. cDNA was synthesized from 500 ng of total RNA using 20 pmol of oligo(dT) and 10 mM dNTP for primary synthesis, and DTT and SuperScript III reverse transcriptase (Invitrogen) for secondary synthesis. The cDNA was amplified by PCR using the following thermal conditions: 10 min at 95 °C, followed by 40–45 cycles of 95 °C for 15 s, and 60 °C for 30 s. All reactions were conducted in duplicate or triplicate. Relative mRNA expression was analyzed by the comparative CT method. The primers used are presented in Table 1.

### 4.3. gDNA Extraction and Mitochondrial DNA Copy Number Analysis

Genomic DNA (gDNA) was extracted from invasive HTR-8/SVneo cells with and without PD-MSC cocultivation using the QIAamp DNA Mini Kit (Qiagen, Valencia, CA, USA). The harvested cells were mixed with 700 μL of lysis buffer containing 20 μg/mL protease K (Cosmo Gene Tech., Seoul, Korea). The samples were then incubated in a 60 °C incubator for 1 h. After incubation, gDNA was extracted following the manufacturer’s procedure. The gDNA concentration was measured with a Nanodrop spectrophotometer (Thermo Scientific, Waltham, MA, USA). gDNA at a concentration of 100 ng/μL was analyzed as discussed by Poidatz et al., [51]. The gene expression of mitochondrial DNA (mtDNA) copy numbers was analyzed by qRT-PCR using 2× TaqMan universal master mix (Applied Biosystems, CA, USA). Nuclear DNA was used as an internal control to calculate the qRT-PCR normalization factor. The primers used are presented in Table 2.

### 4.4. ELISA to Determine ATP Production

HTR-8/SVneo cells (1 × 10^5^/well) were plated with 1.0 mL of serum-free medium in a Trans well system with 6-well inserts (8 μm pore size; Falcon, Corning, NY, USA), and PD-MSCs were plated with 2.0 mL of culture medium in the lower chamber and cultured for 12 h, 24 h and 48 h. The invading cells at the bottom of the insert were harvested with a scraper (Falcon). After the HTR-8/SVneo cells had been harvested, protein lysates were extracted from HTR-8/SVneo cell pellets. The protein concentrations were normalized with a bovine serum albumin protein assay kit (Thermo Fisher, Waltham, MA, USA), and proteins were processed with an ATP production ELISA kit (Thermo Fisher) according to the manufacturer’s recommended procedure.

### 4.5. Invasion Assay

To analyze the invasion ability of trophoblasts with and without PD-MSC cocultivation, HTR-8/SVneo cells (1 × 10^4^/well) were cultured in serum-free medium in 24-well inserts, and PD-MSCs (5 × 10^4^/well) were cultured in cell culture medium containing FBS (Gibco-BRL) in the lower wells of a Trans-well system. Cells in the upper wells were removed with a cotton swab. The invading cells were washed with DPBS (WellGene, Seoul, Korea) and fixed with 100% methanol (Merck, Kenilworth, NJ, USA) for 20 min. The invading cells were stained with Mayer’s hematoxylin (Dako, Carpinteria, CA, USA) for 8 min. The stained cells in 6–9 fields of the membranes were counted under 100× *g* magnification. Each experiment was repeated three times.

### 4.6. Seahorse XF24 Metabolic Flux Analysis

The extracellular acidification rate (ECAR) and oxygen consumption rate (OCR) were determined by using a Seahorse XF24 metabolic flux analyzer (Seahorse Bioscience, North Billerica, MA). PD-MSCs (1.6 × 10^4^/well) were seeded on Seahorse XF24 plates, and HTR-8/SVneo cells (4 × 10^3^/well) were seeded in 24-well inserts (8 μm pore size; Falcon) and incubated for 12 h, 24 h, and 48 h. After the cells had been incubated, cells in the inserts were completely removed with a cotton swab. The invaded HTR-8/SVneo cells had attached to the bottom side of the filter. Occurring on the day of analysis, the film containing invaded HTR-8/SVneo cells was cut, removed, flipped, and placed on a new plate. After a 20-min calibration step in the Seahorse XF24 analyzer, the analysis began with cycles of 2 min of mixing, 2 min of waiting, and 2 min of measurement. The ECAR and OCR were measured at baseline three times before the sequential injection of mitochondrial inhibitors. After each addition of a mitochondrial inhibitor, three measurements were taken before injection of the subsequent inhibitor. The following mitochondrial inhibitors mixed with culture medium at pH 7.4 were used: drug compounds related to glycolysis (glucose: 10 mM, oligomycin: 1.0 μM, 2-DG: 50 mM) and mitochondrial oxygen consumption (oligomycin: 1.0 μM, FCCP: 0.5 μM, rotenone/AA: 0.5 μM). The ECAR and OCR were determined using an XF analyzer (Seahorse Bioscience, North Billerica, MA, USA). All experiments were performed in triplicate.

### 4.7. Statistical Analysis

All experiments were performed at least three times. All data are presented as the means ± standard errors of the means (SEMs). Student’s *t*-test was performed for groupwise comparisons, and a *p*-value less than 0.05 indicated statistical significance.

## 5. Conclusions

To conclude, we demonstrate that PD-MSCs enhance the invasion activity of trophoblasts via MMP-2/9 and the Rho family signaling pathways. Additionally, PD-MSCs trigger altered mitochondrial dynamics (e.g., ROS levels, ATP production, glycolysis, mitochondrial respiration, and calcium signaling) in trophoblasts during invasion. Finally, PD-MSCs promote mitochondrial autophagy in invasive trophoblasts via the PINK1-PARKIN pathway, protecting trophoblasts from mitochondrial damage due to altered metabolism (Figure 6). Our data suggest the principle mechanism of trophoblast invasion via mitochondrial function and a new stem cell therapy for infertility.

## Figures and Tables

**Figure 1 ijms-21-08599-f001:**
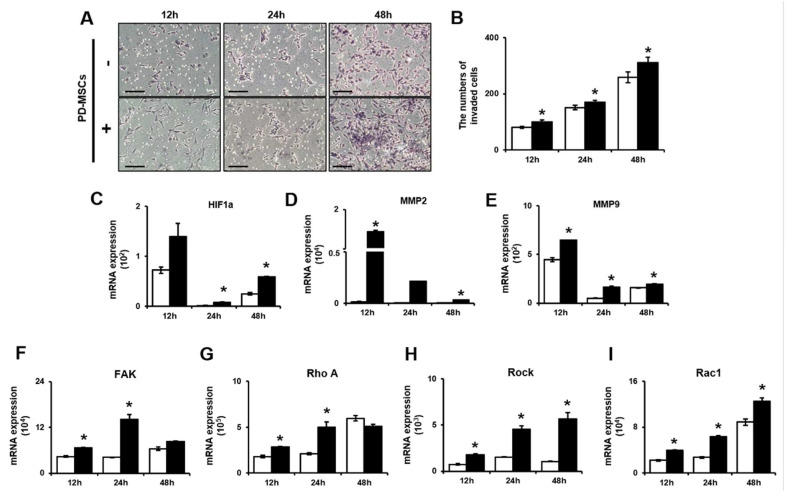
PD-MSCs increased the invasion ability of trophoblasts via activation of MMPs and the Rho family signaling pathway. (**A**) Invasive trophoblasts with and without PD-MSC cocultivation were assessed by invasion assays (magnification, 100× *g*). (**B**) Trophoblasts were quantified by Image J. The mRNA expression of (**C**) HIF1a, (**D**) MMP-2, (**E**) MMP-9, (**F**) FAK, (**G**) Rho A, (**H**) ROCK1 and (**I**) Rac1 in invasive trophoblast cells with and without PD-MSC cocultivation over time. The data was performed at least in triplicate and expressed as the means ± S.D. Indicates * *p* < 0.05. White bar: control, Black bar: PD-MSC cocultivation. *: control vs. PD-MSC cocultivation at each time point. PD-MSCs: Placenta-derived mesenchymal stem cells; MMP: Matrix metalloproteinase; mRNA: messenger RNA; HIF1a: Hypoxia-inducible factor 1a; FAK: Focal adhesion kinase; ROCK1: Rho-associated coiled-coil-containing protein kinase 1; Rac1: Ras-related C3 botulinum toxin substrate 1.

**Figure 2 ijms-21-08599-f002:**
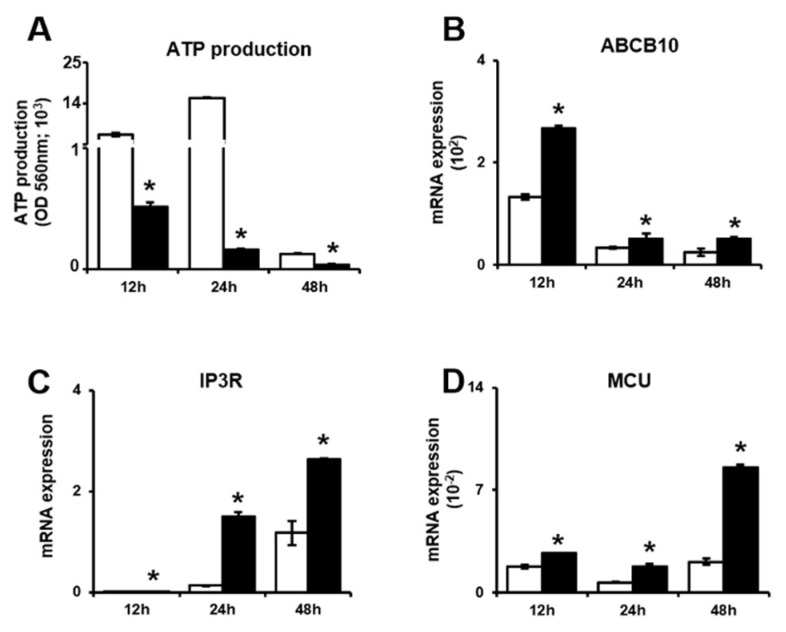
PD-MSCs increased ATP consumption through calcium channel activity in the mitochondria of invasive trophoblasts. (**A**) ATP production in invasive trophoblasts with and without PD-MSC cocultivation was determined by ELISA. The mRNA expression of (**B**) ABCB10, (**C**) IP3R and (**D**) MCU in invasive trophoblast cells with and without PD-MSC cocultivation over time. The data was performed in at least triplicate and expressed as the means ± S.D. Indicates * *p* < 0.05. White bar: control, Black bar: PD-MSC cocultivation. *: control vs. PD-MSC cocultivation at each time point. PD-MSCs: Placenta-derived mesenchymal stem cells; ATP: Adenosine triphosphate; ELISA: Enzyme-linked immunosorbent assay; ABCB10: ATP binding cassette B member 10; IP3R: Inositol trisphosphate receptor; MCU: Mitochondrial calcium uniporter.

**Figure 3 ijms-21-08599-f003:**
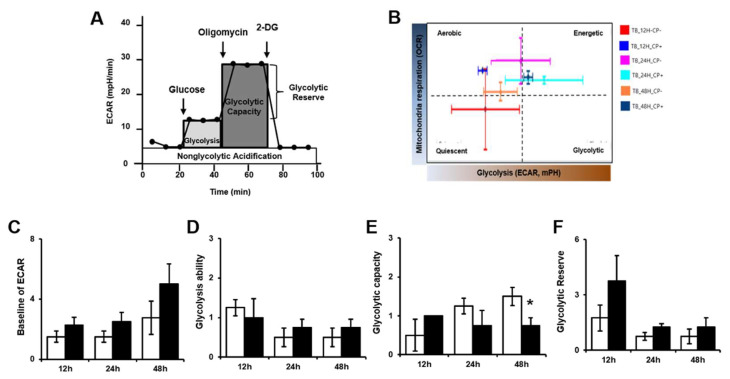
PD-MSCs regulate the glycolytic ability of invasive trophoblasts. (**A**) Schematic illustration of the ECAR. (**B**) The phenotypes of invasive trophoblasts with and without PD-MSC cocultivation over time. (**C**) The baseline ECAR, (**D**) glycolytic ability, (**E**) glycolytic capacity and (**F**) glycolytic reserve of invasive trophoblasts cocultured with PD-MSCs were analyzed by XF assay. The data were performed at least in triplicate and expressed as the means ± S.D. Indicates * *p* < 0.05. White bar: control, Black bar: PD-MSC cocultivation. *: control vs. PD-MSC cocultivation at each time point. PD-MSCs: Placenta-derived mesenchymal stem cells; XF: Extracellular flux.

**Figure 4 ijms-21-08599-f004:**
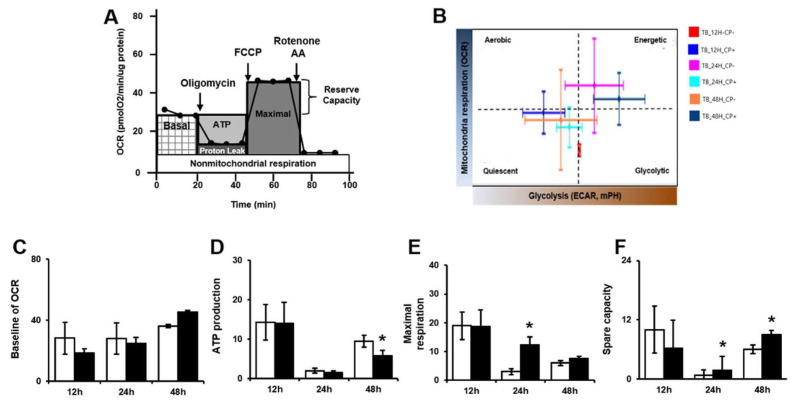
PD-MSCs regulate mitochondrial respiration in invasive trophoblasts. (**A**) Schematic illustration of the OCR. (**B**) The phenotypes of invasive trophoblasts with and without PD-MSC cocultivation over time. (**C**) The baseline OCR, (**D**) ATP production, (**E**) maximal respiration and (**F**) spare capacity of invasive trophoblasts with and without PD-MSC cocultivation were measured by XF assay. The data was performed in at least triplicate and expressed as the means ± S.D. Indicates * *p* < 0.05. White bar: control, Black bar: PD-MSC cocultivation. *: control vs. PD-MSC cocultivation at each time point. PD-MSCs: Placenta-derived mesenchymal stem cells; OCR: Oxygen consumption rate; ATP: Adenosine triphosphate; XF: Extracellular flux.

**Figure 5 ijms-21-08599-f005:**
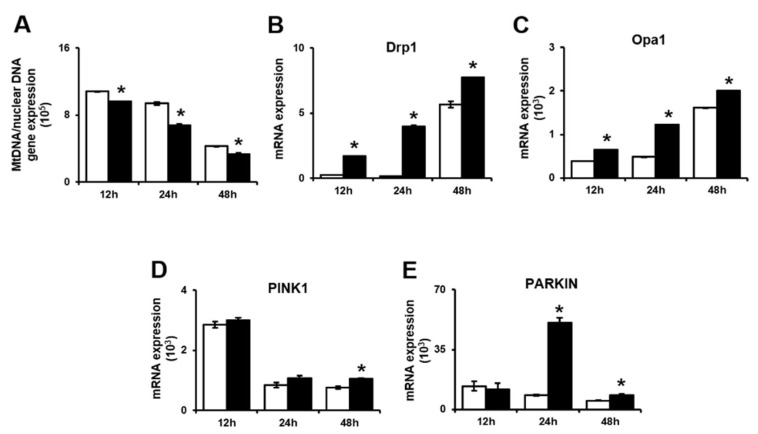
PD-MSCs trigger mitochondrial autophagy in invasive trophoblasts via altering the balance between PINK1 and PARKIN expression. (**A**) The ratio of mitochondrial DNA in gDNA analyzed by qRT-PCR. The mRNA expression of (**B**) Drp1 and (**C**) Opa1 in invasive trophoblast cells with and without PD-MSC cocultivation over time. The mRNA expression of (**D**) PINK1 and (**E**) PARKIN in invasive trophoblast cells with and without PD-MSC cocultivation over time. The data were performed in at least triplicate and expressed as the means ± S.D. Indicates * *p* < 0.05. White bar: control, Black bar: PD-MSC cocultivation. *: control vs. PD-MSC cocultivation at each time point. PD-MSCs: Placenta-derived mesenchymal stem cells; PINK1: PTEN-induced kinase 1; PARKIN: parkin RBR E3 ubiquitin protein ligase; gDNA: Genomic deoxyribonucleic acid; qRT-PCR: Quantitative reverse transcription polymerase chain reaction; mRNA: messenger RNA; Drp1: Dynamin-1-like protein; Opa1: OPA1 mitochondrial dynamin like GTPase.

**Figure 6 ijms-21-08599-f006:**
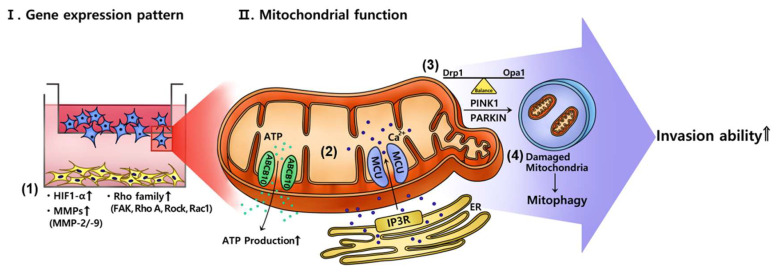
Schematic illustration of the current working hypothesis. The gene expression pattern (**I**) showed that PD-MSCs improve trophoblast invasion via various gene expression. (1) PD-MSCs trigger trophoblast invasion via upregulating HIF1α and activating MMPs and the Rho family signaling pathway. The mitochondrial function (**II**) indicated that PD-MSCs regulate dynamic mitochondrial function including ATP synthesis, calcium transfer and mitophagy in invaded trophoblast cells. (2) PD-MSCs regulate the mitochondrial function of invasive trophoblasts. Particularly, the mRNA expression of both MCU and IP3R, transporters that move calcium ions though the mitochondria and ER, is upregulated in invasive trophoblasts. Additionally, PD-MSCs increase the mRNA expression of ABCB10, which is involved in ATP production, in invasive trophoblasts. (3) PD-MSCs regulate mitochondrial fragmentation via maintaining the balance between fission and fusion in trophoblasts. (4) PD-MSCs trigger mitochondrial autophagy in trophoblasts via upregulating PINK1 and PARKIN expression. Finally, PD-MSCs promote trophoblast invasion via regulating mitochondrial dynamics, including ATP production and mitochondrial autophagy. PD-MSCs: Placenta-derived mesenchymal stem cells; HIF-1a: Hypoxia inducible factor 1 alpha; MMP: Matrix metalloproteinase; ATP: Adenosine triphosphate; mRNA: messenger RNA; MCU: Mitochondrial calcium uniporter; IP3R: Inositol 1,4,5-trisphosphate; ER: Endoplasmic reticulum; ABCB10: ATP binding cassette subfamily B 10; PINK1: PTEN-induced kinase 1; PARKIN: parkin RBR E3 ubiquitin protein ligase

**Table 1 ijms-21-08599-t001:** The primers used to determine the expression of genes by qRT-PCR in this study.

Gene	Forward Primer	Reverse Primer
HIF1a	5’-GTT TAC TAA AGG ACA AGT CA-3’	5’-TTC TGT TTG TTG AAG GGA G-3’
FAK	5’-GGA GCA TTG GGT CGG GAA CTA-3’	5’-CTC-AAT GCA GTT TGG AGG TGC-3’
Rho A	5’-TGG AAA GCA GGT AGA GTT GG-3’	5’-GAC TTC TGG GGT CCA CTT TT-3’
ROCK1	5’-GAT CTT GTA GCT CCC GCA TCT GT-3’	5’-GAA GAA AGA GAA GCT CGA GA-3’
Rac1	5’-TGA TGC AGG CCA TCA AGT GT-3’	5’-AGA ACA CAT CTG TTT GCG GAT AG-3’
MMP2	5’-CGG CCG CAG TGA CGG AAA-3’	5’-CAT CCT GGG ACA GAC GGA AG-3’
MMP9	5’-GAC GCA GAC ATC GTC ATC CAG TTT-3’	5’-GCC GCG CCA TCT GCG TTT-3’
HO-1	5’-TGG TGA TGG CCT CCC TGT ACC ACA TCT-3’	5’-AGA GCT GGA TGT TGA GCA GGA ACG CAG TCT-3’
HO-2	5’-ATG TCA GCG GAA GTG GAA-3’	5’-GGG AGT TTC AGT GCT CGC
SOD1	5’-ATG GCG ACG AAG GCC GTG TGC GTG CTG AAG-3’	5’-TGC CTC TCT TCA TCC TTT GGC-3’
ABCB10	5’-ATG GGC GAT ATC TAC GGA AAC TGA-3’	5’-GGC GAG CTG GAT AGG CAA AAT-3’
IP3R	5’-CGG AGC AGG GTA TTG GAA GGC-3’	5’-GTC CAC TGA GGG CTG AAA CT-3’
MCU	5’-CGT TTC CAG TTG AGA GAT GGC-3’	5’-GAT CCT CTG GTG TAC CGT CC-3’
Drp1	5’-CTG ACG CTT GTG GAT TTA CC-3’	5’-CCC TTC CCA TCA ATA CAT CG-3’
Opa1	5’-GCA GGA TTC AGC AGA TAA-3’	5’-CTC TTC TTC ATA TTC TCT TAT TAG C-3’
PINK1	5’-TGG AGG ATT TAA CCC AGG AG-3’	5’-TTA CCA ATG GAC TGC CCT ATC A-3’
PARKIN	5’-TGG AGG ATT TAA CCC AGG AG-3’	5’-ACA GGG CTT GGT GGT TTT CT-3’
GAPDH	5’-CTC CTC TTC GGC AGC ACA-3’	5’-AAC GCT TCA CCT AAT TTG CGT-3’

**Table 2 ijms-21-08599-t002:** The primers related to mitochondrial DNA copy number used to determine the expression of genes by qRT-PCR in this study.

Gene	Forward Primer	Reverse Primer
Mitochondrial DNA	5’-CCA CTG TAA AGC TAA CTT AGC ATT AAC-3’	5’-GTG ATG AGG AAT AGT GTA AGG AGT ATG G-3’
Nuclear DNA	5’-CCA GAA AAT AAA TCA GAT GGT ATG TAA CA-3’	5’-TGG TTT AGG GTT GCT TCC-3’
Mitochondrial DNA probe	5’-JOE-CCA ACA CCT CTT TAC AGT GAA ATG CCC CA-BHQ1-3’	
Nuclear DNA probe	5’-JOE-CAG CAC TTC TTT TGA GCA CAC GGT CG-BHQ1-3’	

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
