# Peer review of "Mitochondrial Dynamics in Placenta-Derived Mesenchymal Stem Cells Regulate the Invasion Activity of Trophoblast"

_ijms, 2020, doi:10.3390/ijms21228599_

Round 1

Reviewer 1 Report

The authors have conducted an interesting study looking at mitochondrial dynamics in invasive trophoblasts especially during PD-MSC co-cultivation. The study is well written but some of the interpretations can be discussed in more detail. 

  1. When the results are being discussed, it is better to state that a certain change occurs by x fold (depending on your results it may be 1.5 fold or 2 fold or more). When stated like that reader can determine how much of a difference there is better control and experimental cases. 
  2. In Figure 2A, ATP production is low in the invasive trophoblasts co cultivated with PD-MSC according to the graph. So is the graph not labelled correctly? The mRNA expression of ABCB10, IP3R, MCU are increased in invasive trophoblasts but not ATP production?  
  3. In supplement 1 figures, the differences in mRNA expression of HO 1/2 or SOD1 looks very minor /small. The authors should state that the differences are not big and probably discuss the potential reasons for this. 

Author Response

IJMS-957887

Mitochondrial dynamics in placenta-derived mesenchymal stem cells regulate the invasion ability of trophoblast

Jin Seok, Ph.D.course; Sujin Jun. M.S.course; Jung Ok Lee. Ph.D.; Gi Jin Kim. Ph.D.

Reviewer reports:

Reviewer #1 :

Comments and suggestions for authors:

The authors have conducted an interesting study looking at mitochondrial dynamics in invasive trophoblasts especially during PD-MSC co-cultivation. The study is well written but some of the interpretations can be discussed in more detail.

Author’s reply :

We greatly appreciate the reviewer’s positive statement that “The authors have conducted an interesting study looking at mitochondrial dynamics in invasived trophoblasts especially during PD-MSCs co-cultivation.”.

Point #1: When the results are being discussed, it is better to state that a certain change occurs by x fold (depending on your results it may be 1.5 fold or 2 fold or more). When stated like that reader can determine how much of a difference there is better control and experimental cases.

Author’s reply :

Thank you for your comments. We agree with your opinion that When stated like that reader can determine how much of a difference there is better control and experimental cases.” However, most of results are lower control levels compared to those of PD-MSCs cocultivation group. Also, we confirmed the difference of gene expression in trophoblasts cells according to time dependent when they are cocultured with PD-MSCs.

Point #2: In Figure 2A, ATP production is low in the invasive trophoblasts co cultivated with PD-MSC according to the graph. So is the graph not labelled correctly? The mRNA expression of ABCB10, IP3R, MCU are increased in invasive trophoblasts but not ATP production? 

Author’s reply :

Thank you for your comments. In our data, the labelled graph is correct. In figure 2, ATP production is analyzed by whole protein lysate sample of invasived trophoblast, and the mRNA expression of ABCB10, IP3R and MCU are analyzed by RNA sample of invasived trophoblast. As shown in Figure 2, we confirmed decreased ATP production and increased mRNA expression related to ATP production processing in invasived trophoblast.

Point #3: In supplement 1 figures, the differences in mRNA expression of HO 1/2 or SOD1 looks very minor /small. The authors should state that the differences are not big and probably discuss the potential reasons for this.

Author’s reply :

Thank you for your critical comments. As you mentioned, the sentence on the slight difference of expression of HO-1/-2 and SOD1 was corrected in revised manuscripts. Also, it is hard to analyze these ROS and antioxidants related enzyme factors because they are very rapidly dynamics as an intermediator during metabolic process (see below references). Therefore, it should be necessary to establish a system for quantity analysis in order to analyze differences defined expression in the future.

  • Waqar Ahmad, Bushra Ijaz, Khadija Shabbiri, Fayyaz Ahmed and Sidra Rehman et al., Oxidative toxicity in diabetes and Alzheimer’s disease: mechanisms behind ROS/RNS generation. J Biomed Sci. 2017 Sep 19;24(1):76.
  • Veronique Nogueira and Nissim Hay et al., Molecular pathway: Reactive Oxygen Species Homeostasis in cancer cells and implications for cancer therapy. Clin Cancer Res. 2013 Aug 15; 19(16): 4309-4314.
  • Junseong Park, Jungsul Lee and Chulhee Choi et al., Mitochondrial network determines intracellular ROS dynamics and sensitivity to oxidative stress through switching inter-mitochondrial messengers. PLos One. 2011; 6(8):e23211.

Reviewer 2 Report

  • The authors need to perform mitochondrial morphology Visualization (may be with MitoTracker or any suitable assay of author's choice)
  • 1. HTR-8/SVneo cell culture: The authors mentioned that 'HTR-8/SVneo cells (1×105/well) were seeded in the lower chamber and P'. The authors may reconfirm this statement.
  • 5. PD-MSCs trigger mitochondrial autophagy in invasive trophoblasts: The mitochontrial autophagy is well known to be associated with altered levels of nuclear factor-kappa B which in-turn decrease in HTR-8/SVneo invasion. Did the authors studied nuclear factor-kappa B?
  • Also Stomatin-like protein 2 is an important factor which may impair the mitochondrial functions in HTR8/SVneo cells through altered proliferation, invasion and migration. Did the authors studied SLP2?
  • The authors stated that 'high ROS levels inhibit the invasion ability of trophoblasts'. The MMPs (which is increase in the present investigation) is tightly coupled with the ROS modulations. This also needs to be studied.

Author Response

IJMS-957887

Mitochondrial dynamics in placenta-derived mesenchymal stem cells regulate the invasion ability of trophoblast

Jin Seok, Ph.D.course; Sujin Jun. M.S.course; Jung Ok Lee. Ph.D.; Gi Jin Kim. Ph.D.

Reviewer reports:

Reviewer #2

Point #1: The authors need to perform mitochondrial morphology Visualization (may be with MitoTracker or any suitable assay of author's choice)

Author’s reply :

Thank you for your critical comment. We tried to express the mitochondria visualization of invasive trophoblast by MitoTracker. However, the experiments are extremely limited because this study is experimented using insert system for analyzing invasived trophoblast cells. Also, we have a time limitation to analyze the mitochondrial visualization of trophoblast cells according to PD-MSCs cocultivation due to COVID-19 pandemic situation. However, further studies are needed in the future as you commented.

Point #2: 1. HTR-8/SVneo cell culture: The authors mentioned that 'HTR-8/SVneo cells (1×105/well) were seeded in the lower chamber and P'. The authors may reconfirm this statement. We modified this part in revised manuscripts.

Author’s reply :

Thank you for your comment. In our study, we used insert system for indirect co-cultivation between PD-MSCs and HTR-8/SVneo cells. There was a mistake in our labeling in method part of manuscripts. We corrected the mistakesentences in revised manuscripts and highlighted it. See figure below.

Figure. Simplified experimental scheme.

Point #3: 5. PD-MSCs trigger mitochondrial autophagy in invasive trophoblasts: The mitochontrial autophagy is well known to be associated with altered levels of nuclear factor-kappa B which in-turn decrease in HTR-8/SVneo invasion. Did the authors studied nuclear factor-kappa B?

Also Stomatin-like protein 2 is an important factor which may impair the mitochondrial functions in HTR8/SVneo cells through altered proliferation, invasion and migration. Did the authors studied SLP2?

Author’s reply :

Thank you for your comments. To analyze mitochondrial autophagy, we analyzed the expressions of PINK1 and PARKIN, which are widely known mitochondrial autophagy regulators gene in invasive trophoblast. Also, we analyzed mRNA expression related to ATP generation including ABCB10, IP3R and MCU in invasived trophoblast. Our main objective is to study the effects of PD-MSCs on trophoblast invasion ability, and the influence of PD-MSCs cocultivation on mitochondrial function invasived trophoblast. So, we focused on mitochondrial function for ATP production in trophoblast cells according to PD-MSCs cocultivation.

Of course, NF-kB and Stomatin-like protein 2 you recommended are important factors for studying on mitochontrial autophagy. However, we have several limitations to analyze NFkB, the autophagy regulator and SLP2, mitochondrial function regulator in invasive trophoblast cells according to PD-MSCs cocultivation due to COVID-19 pandemic situation. So, we added the relevant references in revised manuscripts, and further studies are needed in the future as you commented.

  • Soo-Young Oh, Jae Ryoung Hwang, Minji Choi, Yoo-Min Kim, Jung-Sun Kim, Yeon-Lim Suh, Suk-Joo Choi and Cheong-Rae Roh et al., Autophagy regulates trophoblast invasion by targeting NF-kB activity. Sci Rep. 2020 Aug 20;10(1):14033.
  • Hadia Moindjie, Esther Dos Santos, Rita-Josiane Gouesse, Nelly Swierkowski-Blanchard, Valérie Serazin, Eytan R Barnea, François Vialard and Marie-Noëlle Dieudonné et al., Preimplantation factor is an anti-apoptotic effector in human trophoblasts involving p53 signaling pathway. Cell Death Dis. 2016 Dec 1;7(12)e2504.
  • Xue Zhang, Bing-Yi Li, Li-Juan Fu, Enoch Appiah Adu-Gyamfi, Bai-Ruo Xu, Tai-Hang Liu, Xue-Mei Chen, Xi Lan, Ying-Xiong Wang, Hong-Bing Xu and Yu-Bin Ding et al., Stomatin-like protein 2 (SLP2) regulates the proliferation and invasion of trophoblast cells by modulating mitochondrial functions. Placenta. 2020 Oct;100:13-23.

Point #4: The authors stated that 'high ROS levels inhibit the invasion ability of trophoblasts'. The MMPs (which is increase in the present investigation) is tightly coupled with the ROS modulations. This also needs to be studied.

Author’s reply :

Thank you for your critical comments. We mentioned that “high ROS levels inhibit the invasion ability of trophoblasts.”. Kim et al and their colleagues reported that moderate ROS levels support cell proliferation and migration. However, excessive ROS accumulation promotes sever cellular damage and triggers apoptosis (Kim et al, 2016, Exp Mol Med.; Svineng et al, 2008 Connect Tissue Res.). This result cannot be shown because it is currently under revision, we analyzed the proliferation and cell death of trophoblast according to PD-MSCs cocultivation by trypan blue staining and FACS analysis. As a results, PD-MSCs cocultivation did not affect apoptosis of trophoblast cells.

Additionally, we confirmed that mRNA expressions of MMP-2/-9 were increased in trophoblast with PD-MSCs cocultivation compared to trophoblasts cultured alone. Also, we analyzed MMP-2/-9 activities in supernatant by zymography. As a results, MMP-2/-9 expression in supernatants were increased in trophoblast cocultured with PD-MSCs compared to trophoblast cultured alone. These data attached in revised supplementary figure files.

As your mentioned that “The MMPs is tightly coupled with the ROS modulations”, it is important, In previous reports, Mori et al., and their colleagues reported that mitochondrial ROS signaling increases MMP9 mRNA stability and affects invasiveness cells (Mori et al, 2019, FEBS J; Kar et al, 2010 Physiol Neurobiol). Also, in the present  study, we demonstrated PD-MSCs cocultivation enhance the invasion ability as well as mitochondrial function of trophoblast. Based on these data, we mentioned the correlation between MMPs and ROS levels in revised paper. (please check below) Also, we have several limitations to analyze direct correlationship between MMPs and ROS levels in trophoblast cells regardless PD-MSCs due to COVID-19 pandemic situation. However, further studies are needed in the future as you commented.

Figure. PD-MSCs cocultivation induced MMP-2/-9 activities in trophoblast cells.

  • Jongdoo Kim, Jaehong Kim and Jong-Sup Bae et al., ROS homeostasis and metabolism: a critical liaison for cancer therapy. Exp Mol Med. 2016 Nov 4; 48(11):e269.
  • Gunbjørg Svineng, Chandra Ravuri, Oddveig Rikardsen, Nils-Erik Huseby and Jan-Olof Winberg et al., The role of reactive oxygen species in integrin and matrix metalloproteinase expression and function. Connect Tissue Res. 2008;49(3):197-202.
  • Kazunori Mori, Tetsu Uchida, Toshihiko Yoshie, Yuko Mizote, Fumihiro Ishikawa, Masato Katsuyama and Motoko Shibanuma et al., A mitochondrial ROS pathway controls matrix metalloproteinase 9 levels and invasive properties in RAS-activated cancer cell. FEBS J. 2019 Feb;286(3):459-478.
  • Supriya Kar, Sita Subbaram, Pauline M. Carrico, and J. Andrés Melendeza et al., Redox-control of Matrix Metalloproteinase-1: A critical link between free radicals, matrix remodeling and degenerative disease. Respir Physiol Neurobiol. 2010 Dec 31; 174(3):299-306.

Reviewer 3 Report

Study by Seok et al, provides the insight into how mitochondrial dynamics plays role in placenta-derived mesenchymal stem cells in regulating invasion activity of trophoblasts.

Overall the paper is poorly written with lack of references for their points. for example. line 88, authors tell '' Hypoxia inducible factor 1 alpha (HIF1α) is a transcription factor known to regulate cell migration89 as well as cellular metabolism'' where is the citation for this statement?In fig1,  ''PD-MSCs increased the invasion ability of trophoblasts via activation of MMPs and the Rho family signaling pathway'' ... where is the conclusive data for the activation of MMPs?Authors should show the MMP activities by zymography while putting statement like activity.Authors are missing protein expression data for FAK, Rho expression to actually show the involvement of the pathway.

Author Response

IJMS-957887

Mitochondrial dynamics in placenta-derived mesenchymal stem cells regulate the invasion ability of trophoblast

Jin Seok, Ph.D.course; Sujin Jun. M.S.course; Jung Ok Lee. Ph.D.; Gi Jin Kim. Ph.D.

Reviewer reports:

Reviewer #3

Point #1: Study by Seok et al, provides the insight into how mitochondrial dynamics plays role in placenta-derived mesenchymal stem cells in regulating invasion activity of trophoblasts.

Overall the paper is poorly written with lack of references for their points. for example. line 88, authors tell '' Hypoxia inducible factor 1 alpha (HIF1α) is a transcription factor known to regulate cell migration as well as cellular metabolism'' where is the citation for this statement?

Author’s reply :

Thank you for your comments.
As your mentioned “Hypoxia inducible factor 1 alpha (HIF1α) is a transcription factor known to regulate cell migration~ ”, we added references in revised manuscripts (See below references). Also, we carefully checked references in out manuscript you requested.

References

  • Amanda R Highet, Sultana M Khoda, Sam Buckberry, Shalem Leemaqz, Tina Bianco-Miotto, Elaine Harrington, Carmela Ricciardelli and Claire T Roberts et al. Hypoxia induced HIF-1/HIF-2 activity alters trophoblast transcriptional regulation and promotes invasion. Eur J Cell Biol. 2015 Dec;94(12):589-602.
  • Ayako Nagao, Minoru Kobayashi, Sho Koyasu, Christalle C. T. Chow and Hiroshi Harada et al. HIF-1-dependent reprogramming of glucose metabolic pathway of cancer cells and its therapeutic significance. Int J Mol Sci. 2019 Jan; 20(2):238.
  • Manuel J Del Rey, Álvaro Valín, Alicia Usategui, Carmen M García-Herrero, María Sánchez-Aragó, José M Cuezva, María Galindo, Beatriz Bravo, Juan D Cañete, Francisco J Blanco, Gabriel Criado, José L Pablos et al. Hif-1α knockdown reduces glycolytic metabolism and induces cell death of human synovial fibroblasts under normoxic conditions.Sci Rep. 2017 Jun 16;7(1):36444.

Point #2:

In fig1, ''PD-MSCs increased the invasion ability of trophoblasts via activation of MMPs and the Rho family signaling pathway'' ... where is the conclusive data for the activation of MMPs? Authors should show the MMP activities by zymography while putting statement like activity.

Author’s reply :

Thank you for your comments. In case of the MMPs activities, we added these data in supplementary figure 2. You can check the detail response to the opinion of Reviewer 2.

Point #3:

Authors are missing protein expression data for FAK, Rho expression to actually show the involvement of the pathway.

Author’s reply :

Thank you for your comments.

As mentioned the Materials and Methods in manuscript, we analyzed the mRNA expressions for FAK and Rho in invasived trophoblasts regardless of PD-MSCs cocultivation. It was difficult to analyze protein expression using Western blot in insert-based system although we tried to analyze their protein expressions. However, further studies should be needed in the future as you commented.

Round 2

Reviewer 3 Report

No further comments